genomics, evolution, bioinformatics

X chromosome inactivation, marsupial, DNA methylation, *Rsx*, koala, whole-genome bisulfite sequencing

**Authors for correspondence:**
David Alvarez-Ponce
e-mail: dap@unr.edu
Soojin V. Yi
e-mail: soojin.yi@biology.gatech.edu

# Koala methylomes reveal divergent and conserved DNA methylation signatures of X chromosome regulation

Devika Singh[1], Dan Sun[1], Andrew G. King[2], David E. Alquezar-Planas[2], Rebecca N. Johnson[2,3], David Alvarez-Ponce[4] and Soojin V. Yi[1]

[1]School of Biological Sciences, Georgia Institute of Technology, Atlanta, GA, USA
[2]Australian Museum Research Institute, Australian Museum, Sydney, New South Wales, Australia
[3]National Museum of Natural History, Smithsonian Institution, Washington, DC, USA
[4]Department of Biology, University of Nevada Reno, Reno, NV, USA

DSi, 0000-0001-9560-2079; DS, 0000-0003-2147-9308; AGK, 0000-0003-0481-3490;
DEA-P, 0000-0001-5360-5263; RNJ, 0000-0003-3035-2827; DA-P, 0000-0002-8729-1036;
SVY, 0000-0003-1497-1871

X chromosome inactivation (XCI) mediated by differential DNA methylation between sexes is an iconic example of epigenetic regulation. Although XCI is shared between eutherians and marsupials, the role of DNA methylation in marsupial XCI remains contested. Here, we examine genome-wide signatures of DNA methylation across fives tissues from a male and female koala (*Phascolarctos cinereus*), and present the first whole-genome, multi-tissue marsupial 'methylome atlas'. Using these novel data, we elucidate divergent versus common features of representative marsupial and eutherian DNA methylation. First, tissue-specific differential DNA methylation in koalas primarily occurs in gene bodies. Second, females show significant global reduction (hypomethylation) of X chromosome DNA methylation compared to males. We show that this pattern is also observed in eutherians. Third, on average, promoter DNA methylation shows little difference between male and female koala X chromosomes, a pattern distinct from that of eutherians. Fourth, the sex-specific DNA methylation landscape upstream of *Rsx*, the primary *lnc*RNA associated with marsupial XCI, is consistent with the epigenetic regulation of female-specific (and presumably inactive X chromosome-specific) expression. Finally, we use the prominent female X chromosome hypomethylation and classify 98 previously unplaced scaffolds as X-linked, contributing an additional 14.6 Mb (21.5%) to genomic data annotated as the koala X chromosome. Our work demonstrates evolutionarily divergent pathways leading to functionally conserved patterns of XCI in two deep branches of mammals.

## 1. Introduction

X chromosome inactivation (XCI) is a classic example of sex chromosome regulation in which one of the two X chromosomes in females is silenced as a mechanism thought to adjust the expression levels of X-linked genes [1]. Although XCI is observed in the two deep branches of mammals—eutherian and marsupial mammals [2]—there are several notable differences between the two lineages. First, in eutherians, the transcription of a long non-coding RNA (*lnc*RNA) gene, *Xist*, from the inactive X chromosome is essential for XCI [3–5]. However, the *Xist* locus is not present in marsupials [6,7]. Instead, another *lnc*RNA gene, *Rsx*, drives marsupial XCI [8]. Second, marsupials exhibit 'imprinted' XCI by selectively silencing the paternal X chromosome [9,10]. By contrast, XCI in eutherians occurs randomly between the maternally and paternally derived X chromosomes, although paternal XCI has been observed during early rodent development [11,12]. Third, while eutherian XCI involves the exclusion of active histone

marks and the recruitment of repressive histone marks on the inactive X chromosome [13], marsupial X chromosomes do not show a consistent pattern [10,14]. Instead, the inactive marsupial X chromosome, while depleted of the active histone marks, shows variable enrichment patterns of repressive histone marks [10,14]. Specifically, out of five repressive marks examined in two marsupial studies, H3K9me3, H4K20me3 and HP1α were enriched [14], while H3K27me3 and H3K9me2 [15] were not enriched, on the inactive X chromosome. These differences suggest that evolutionary pathways leading to XCI probably differ between eutherians and marsupials, and that novel insights into the mechanism of XCI can be gained from comparative studies.

The role of DNA methylation in marsupial XCI has been particularly controversial. Immunofluorescent labelling studies observed relative hypomethylation of the inactivate X chromosome in marsupials [15,16]. Other studies found little difference in DNA methylation between active and inactive marsupial X chromosomes [10,17,18]. Recently, Waters et al. [19] analysed reduced representation bisulfite sequencing (RRBS) data of a male and female opossum (Monodelphis domestica) and proposed that female X chromosomes in marsupials, but not in eutherians, exhibit hypomethylation near the transcription start sites (TSSs). Notably, all these studies analysed different marsupial species and tissues. In addition, and importantly, they either examined a small number of CpGs or employed methodologies that over-represent promoters and CpG islands (in the case of RRBS [20]). Since patterns of DNA methylation vary greatly among distinctive genomic regions with different functional consequences, it is necessary to extend our knowledge to unbiased, whole-genome assays of DNA methylation.

Recently, Johnson et al. [21] integrated long and short read sequencing by PacBio and Illumina to generate the highest quality reference genome assembly of any marsupial species for the modern koala (Phascolarctos cinereus), the sole extant member of the marsupial family Phascolarctidae [22]. To leverage and compliment this resource, here we have generated whole-genome bisulfite sequencing (WGBS) maps across tissues of both sexes, capturing the DNA methylation state of nearly all cytosines in koala genome. Our data provide the first multi-tissue, whole-genome methylome resource of any marsupial enabling us to show distinctive impacts of DNA methylation on tissue-specific gene expression in marsupials, as well as on XCI in eutherians and marsupials.

## 2. Results

### (a) Genome-wide differential DNA methylation between tissues in the modern koala

To investigate genome-wide patterns of DNA methylation, we generated WGBS data from five tissues (brain, lung, kidney, skeletal muscle and pancreas) from a male ('Ben', Australian Museum registration M.45022) and a female koala ('Pacific Chocolate', Australian Museum registration M.47723). The mean depth of coverage fell between 9.9× and 14.6× (electronic supplementary material, table S1). The overall DNA methylation levels of koala tissues are on par with those in other mammals [23–25], exhibiting heavy genome-wide DNA methylation punctuated by the hypomethylation of CpG islands and other regulatory elements (figure 1). A hierarchical clustering of methylation profiles demonstrated a clear grouping of samples by tissue (figure 1a). Interestingly, we observed that the pancreas exhibited the most unique methylation signature among the five tissues studied, while the kidney and lung samples shared the most similar methylation profiles.

To further examine patterns of tissue-differential DNA methylation, we identified shared and tissue-specific differentially methylated regions (DMRs) using BSmooth [26]. Tissue-specific DMRs were defined as regions that were differentially methylated in a particular tissue compared to all other tissues in a pairwise analysis, while shared DMRs were those observed in multiple tissues (figure 1b). We found that the majority (50–53%) of tissue-specific DMRs fell in gene bodies (figure 1c; electronic supplementary material, figure S1 and table S2), which was a significant increase compared to length and GC-matched control regions (fold enrichment (FE) = 1.25–1.44, $p < 0.0001$ based on 10 000 bootstraps; figure 1c; electronic supplementary material, figure S1 and table S2). On the other hand, DMRs were significantly depleted in intergenic regions ($p < 0.05$ based on 10 000 bootstraps; figure 1c; electronic supplementary material, figure S1 and table S2).

The numbers of DMRs per tissue are shown in figure 1b. Interestingly, the pancreas samples contained the largest number of tissue-specific DMRs (figure 1b). Further analysis with a more comprehensive sampling of tissues is required to determine if the pancreas is a true outlier in terms of DNA methylation in this species. However, it is worthwhile to note that koalas are known for their unique and highly specialized diet of eucalyptus leaves, which is highly toxic to most other mammals [27]. Indeed, we found that genes containing tissue-specific DMRs (e.g. figure 1d) were enriched in specific biological functions, consistent with their unique tissue origins (electronic supplementary material, table S3). For example, pancreas-specific DMRs were preferentially found in genes associated with metabolic processes while brain-specific DMRs were linked to genes associated with neural developmental processes.

### (b) Global patterns of DNA methylation and transcription in koalas

To infer the role of DNA methylation in gene expression, we integrated methylome data with previously generated koala RNA-seq data [28], identifying matched sets for three common tissues (kidney, brain and lung). Promoter DNA methylation and gene expression were significantly negatively correlated across the genome (table 1; electronic supplementary material, figure S2). In comparison, both extremely hypomethylated and hypermethylated gene bodies showed high gene expression (table 1; electronic supplementary material, figure S2), which is consistent with the patterns observed in other taxa [29–32]. Next, we compared differentially methylated genes (DMGs) containing DMRs ($n = 1944$ genes from $n = 4615$ DMRs) with differentially expressed genes (DEGs), between brain and kidney samples. Currently available RNA-seq data from koalas do not include sufficient biological replicates. We overcame this limitation by simulating replicates within each RNA-seq dataset (NOISeq [33]) and identified 600 putative DEGs (probability of differential expression greater than 95% according to the NOISeq).

DMGs were significantly more likely to be differentially expressed than non-DMGs, exhibiting a 1.54-fold enrichment

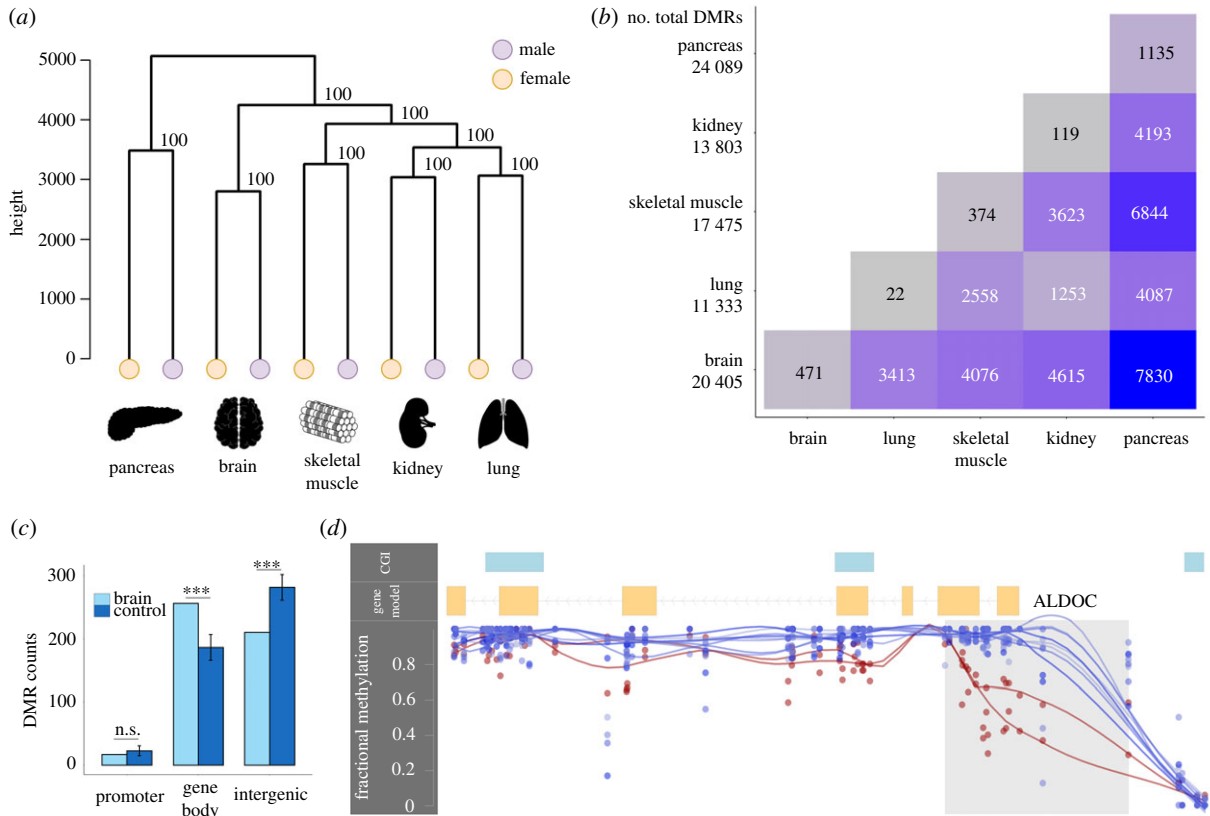

**Figure 1.** Overview of DNA methylation patterns across the koala genome. (a) Hierarchical clustering of DNA methylation of five tissues. (b) Tissue-specific and shared differentially methylation regions (DMRs) between tissues. Total DMRs per tissue are reported in the first column. (c) Enrichment of brain DMRs in different functional regions compared to length and GC-matched control regions (***$p < 0.0001$, n.s., not significant, from 10 000 bootstraps). Error bars depict standard deviation. Results for other tissues are in electronic supplementary material, figure S1. (d) A 945 bp brain-specific DMR overlapping *ALDOC*'s promoter and part of the gene body (shaded region) with corresponding CpG fractional methylation for the brain (lower lines) and eight remaining tissues (top lines). Line smoothing performed using local regression (LOESS). This gene was upregulated in the brain compared to the kidney (probability of differential expression greater than 96% from NOISeq).

**Table 1.** Correlation analysis of mean promoter and gene body DNA methylation and ranked gene expression. Spearman's rank correlation coefficients ($\rho$) and associated significances are reported for all tissues with both WGBS data and RNA-seq expression data.

| tissue | genomic region | gene count | $\rho$ (*p*-value) |
|---|---|---|---|
| brain | promoter | 5396 | $-0.08$ ($p = 2.28 \times 10^{-9}$) |
| | gene body | 5443 | $-0.16$ ($p < 2.2 \times 10^{-16}$) |
| kidney | promoter | 9268 | $-0.12$ ($p < 2.2 \times 10^{-16}$) |
| | gene body | 9379 | $-0.12$ ($p < 2.2 \times 10^{-16}$) |
| lung | promoter | 9192 | $-0.13$ ($p < 2.2 \times 10^{-16}$) |
| | gene body | 9265 | $-0.19$ ($p < 2.2 \times 10^{-16}$) |

($\chi^2 = 33.07$, $p < 0.0001$). Additionally, differential expression between tissues displayed a weak, yet significant negative correlation with differential promoter DNA methylation between tissues (electronic supplementary material, figure 3A). Gene body DNA methylation showed a more complex relationship with gene expression where both relative hypo- and hypermethylation was associated with increased expression (electronic supplementary material, figure S3B). These results indicate significant associations between DNA methylation and transcription in the koala genome, where the direction

of relationship is consistent with previous observations in other taxa [29–32].

## (c) Global hypomethylation of female X chromosome in koalas

Using the novel WGBS data from both sexes in koalas, we examined variations in male and female X chromosome DNA methylation. The koala genome project used cross-species chromosome painting data to identify 24 putative X chromosome scaffolds and 406 putative autosomal scaffolds [21]. As expected from 2:1 ratio of X chromosomes in females compared to males, the median depth of coverage of CpGs on the putative X scaffolds were consistently higher (approx. twofold) in female samples compared to male samples ($p < 2.2 \times 10^{-16}$, Mann–Whitney $U$-test; electronic supplementary material, figure S4A). Furthermore, the proportion of reads mapped to the putative X scaffolds showed a distinct bimodal distribution whereby the male samples cluster close to 1.3% and the female samples cluster near 2.4% (electronic supplementary material, figure S4B). By contrast, male and female samples were indistinguishable with respect to read mapping to putative autosomes (electronic supplementary material, figure S4D). These observations demonstrate that our WGBS data are well suited to study differential DNA methylation between the male and female X chromosomes.

We found that the global DNA methylation level of the female X chromosome was strikingly lower than that of

*Proc. R. Soc. B* **288**: 20202244

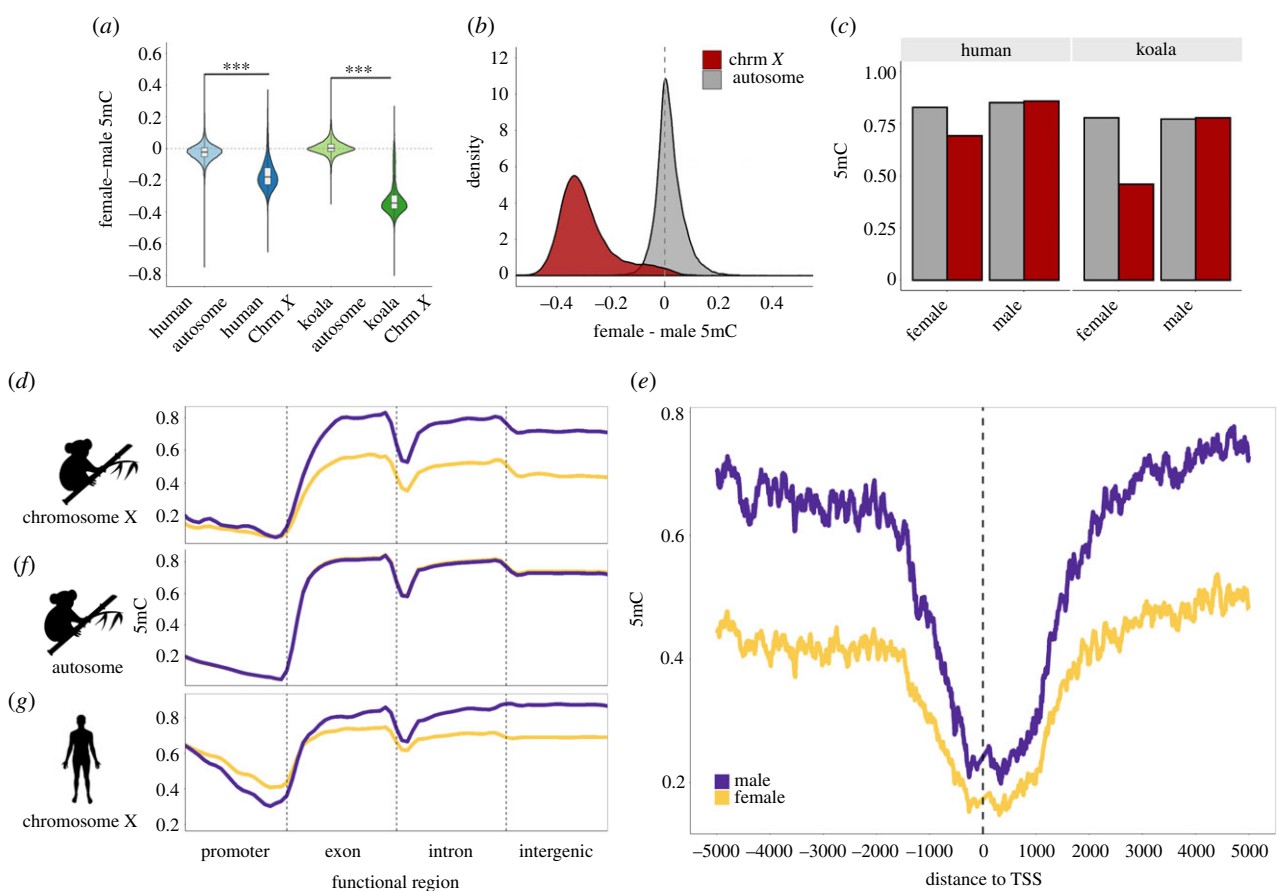

**Figure 2.** Global patterns of female and male DNA methylation (5mC) in human and koala X chromosomes. (*a*) Both human and koala X chromosomes show chromosome-wide female hypomethylation. (*b*) Distributions of the DNA methylation difference between female and male koalas in autosomes and the X chromosome. (*c*) Comparison of fractional DNA methylation between females and males illustrating that the female X chromosomes are hypomethylated in both humans and koalas. (*d*, *f* and *g*) DNA methylation differences between females and males of (*d*) the X chromosome and (*f*) autosomes of koalas and (*g*) the X chromosome of female (yellow) and male (purple) humans. In the koala autosome, the female and male lines overlap completely illustrating no sex-based methylation difference. Line smoothing was performed using local regression (LOESS). (*e*) Average fractional methylation of CpGs in 100 bp sliding windows using a 10 bp step size in a 5 kb region upstream and downstream of all chromosome X-linked gene's transcription start sites (TSSs) across koala tissues.

the male X chromosome in all koala tissues examined (figure 2*a*, *b*; electronic supplementary material, figure S5; $p < 2.2 \times 10^{-16}$, Mann–Whitney *U*-test). This trend could either be attributed to the reduction of DNA methylation in the female X chromosomes or the increase of DNA methylation in the male X chromosome. We compared the male and female DNA methylation for autosomes and determined that the female X chromosome exhibited reduced DNA methylation (figure 2*c*). Consequently, we use the term 'female hypomethylation' (as opposed to male hypermethylation) consistently in this work. We also analysed DNA methylation of human male and female X chromosomes (Methods) and found that the human X chromosomes were also globally hypomethylated in females compared to males (figure 2*a*,*c*).

Significant female hypomethylation was observed in all functional regions across the koala X chromosome (figure 2*d*; electronic supplementary material, figure S6A) but was the most pronounced in gene bodies and intergenic regions. Promoters showed the least sex-based DNA methylation difference. In figure 2*e*, we show a zoomed-in view of the male and female X chromosome DNA methylation near the TSS, which illustrates the clear pattern of consistent female hypomethylation. The koala autosomal scaffolds, on the other hand, did not display significant differential DNA methylation between the sexes in any functional region (figure 2*f*; electronic supplementary material, figure S6B).

In comparison, female X chromosome hypomethylation in humans (figure 2*a*,*c*; electronic supplementary material, figure S6C) was driven by the gene body and intergenic regions, while promoters displayed female hypermethylation (figure 2*g*).

## (d) Promoter DNA methylation is not a universal driver of sex-specific expression in koalas

To investigate the implications of the observed sex-specific DNA methylation, we examined sex-specific expression using published RNA-seq koala transcriptomes [28]. Of the total RNA-seq dataset, only one tissue (kidney) had expression data from both sexes, and was used for downstream analysis. Of the 209 X-linked genes, 36 (17.2%) exhibited female overexpression while 11 (5.3%) showed male overexpression (probability of differential expression greater than 95% based on NOISeq; figure 3*a*). Although, on average, autosomal genes also exhibited slight female-biased expression (electronic supplementary material, figure S7A,B), the increase was more substantial in the X chromosome (mean chromosome X female to male $\log_2$ fold change = 0.50, autosome female to male expression $\log_2$ fold change = 0.24).

We examined the relationship between fractional methylation difference and gene expression difference between males and females ($n = 209$ gene bodies and $n = 206$

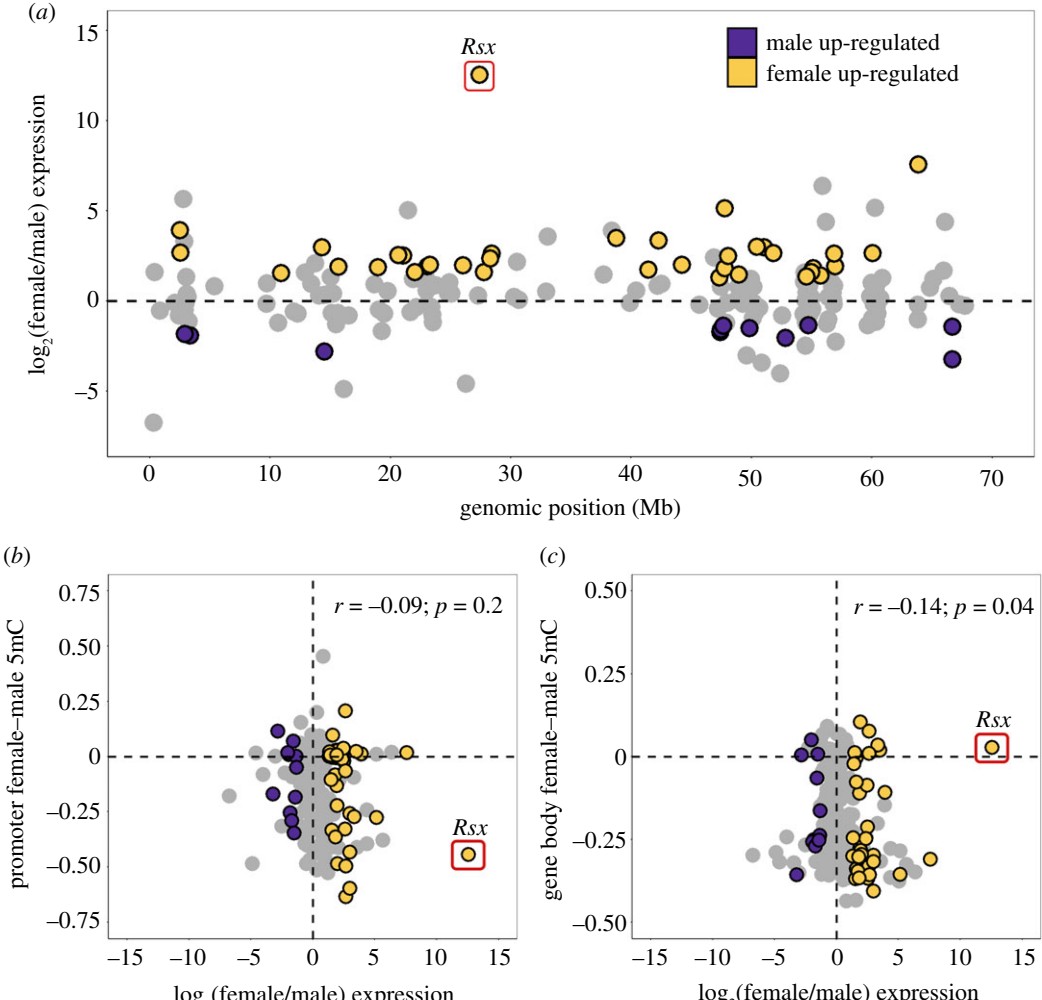

**Figure 3.** Female and male gene expression across autosomes and the X chromosome using kidney RNA-seq data. (*a*) Distribution of female (yellow) and male (purple) upregulated genes by NOISeq (probability of differential expression greater than 95%) across the X chromosome (scaffolds ordered by scaffold length). (*b,c*) The mean female and male fractional methylation difference across each gene promoter (*b*) and gene body (*c*) correlated with the corresponding log-transformed ratio of female to male expression. Spearman's rank correlation coefficient and the associated *p*-value are reported. The *Rsx* gene was excluded from the correlation calculation.

promoters, excluding three promoters with CpGs coverage less than 3). In promoters, no significant relationship was observed (figure 3*b*). Indeed, both hypo- and hypermethylated promoters were similarly represented in female over-expressed genes (electronic supplementary material, table S4). Interestingly, female and male DNA methylation difference in gene bodies showed a significant negative correlation with gene expression (Spearman's rank correlation coefficient, $\rho = -0.14$, $p = 0.04$; figure 3*c*). These observations support an association between sex-based differential gene body DNA methylation and differential gene expression in koalas.

### (e) The *Rsx* region displays a pattern suggesting methylation-driven control of X chromosome regulation in koalas

We sought to infer the role of DNA methylation on the main driver gene of marsupial XCI. Previous studies have indicated that *Rsx*, a key regulator of XCI, is regulated by sex-specific DNA methylation in the opossum [8,10]. To examine if the koala *Rsx* also exhibits regulatory signatures of differential DNA methylation, we first identified the putative *Rsx* region from this species. Based on the sequence homology

with the *Rsx* gene from the grey short-tailed opossum (*Monodelphis domestica*) [8], we identified a 29.8 kb candidate *Rsx* sequence (Methods), using PacBio long read sequencing generated by Johnson *et al*. [21]. We validated that the candidate *Rsx* in koala was significantly upregulated in females compared to males across different tissues, using two different tools to measure differential gene expression (table 2).

We found that the gene body region of *Rsx* is similarly methylated between the male and female koalas (figure 4; see also figure 3*c*). However, two CpG islands upstream of *Rsx* are highly and significantly female hypomethylated. Specifically, these CpG islands covering 101 CpGs exhibited a 36% reduction of DNA methylation in females compared to males (figure 4). These observations indicate that differential expression of koala *Rsx* between sexes is likely under the regulation of differential DNA methylation of upstream *cis*-regulatory sequences.

### (f) Identification of novel candidate X-linked scaffolds by sex-specific methylation patterns

We have demonstrated above (section 2a) several characteristics of the X-linked scaffolds that distinguished them from autosomal scaffolds. Specifically, we showed that

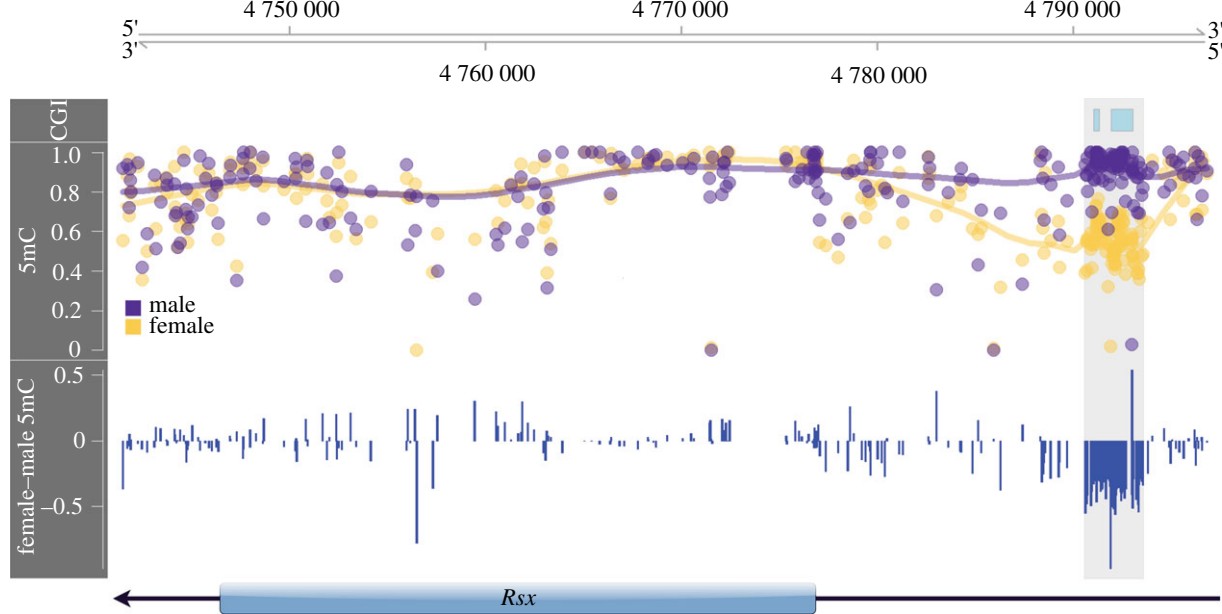

**Figure 4.** Annotation of genomic of DNA methylation (5mC) around *Rsx*. The top panel identifies CpG islands (CGI), the middle panel reports the absolute male (purple) and female (yellow) fractional methylation at each CpG, and the bottom panel shows the female and male fractional methylation difference. Highlighted in grey across all panels is the female hypomethylated region upstream of the *Rsx* TSS.

**Table 2.** Sex-based differential expression of the *lnc*RNA *Rsx* using different data subsets and expression quantification tools. Normalized expression count values and significance of sex-based differential expression is shown for three data subsets using two expression quantification tools. All data refer to the dataset considering all 15 RNA-seq samples (7 male and 8 female). Matched data include the tissues with both male and female RNA-seq samples (brain, kidney and lung), and the kidney data are reported independently. DeSeq2 reports significance as an associated *p*-value from the Wald test, while NOISeq reports a probability of differential expression threshold.

| expression dataset | tool | female count | male count | significance |
| --- | --- | --- | --- | --- |
| all data ($n = 15$) | DeSeq2 | 6987.1 | 16 | $p$-value $= 0.05$ |
| matched data ($n = 6$) | DeSeq2 | 6837.6 | 0 | $p$-value $= 2.04 \times 10^{-30}$ |
| matched data ($n = 6$) | NOISeq | 7872.4 | 0.67 | probability $= 99.99\%$ |
| kidney data ($n = 2$) | NOISeq | 4074.4 | 0.68 | probability $= 99.99\%$ |

X-linked scaffolds exhibited significantly higher sequence depths in females than in males, distinctive clustering based on the proportion of mapped reads in males and females and distinctive hypomethylation in females compared to males (electronic supplementary material, figure S4). We used these characteristics to determine if additional candidate X scaffolds existed within the 6.7% of the koala assembly that remained unclassified. We identified 98 scaffolds that fit the above patterns (electronic supplementary material, figure 4C), including a clear shift towards female hypomethylation (mean female–male 5mC for all candidate X scaffolds was −0.25 ± 0.12) (electronic supplementary material, tables S5 and S6). These candidate scaffolds contributed an additional 14.6 Mb (21.5%) to the annotated koala X chromosome. These newly identified putative X chromosome scaffolds should further our understanding of the koala X chromosome.

## 3. Discussion

Whole-genome bisulfite sequencing is a gold-standard of genomic DNA methylation analysis, as it produces information on nearly all cytosines in a genome. We generated WGBS data from a male and female koala, including the same individual whose genome was recently sequenced to yield the highest quality assembly among current marsupial genomes [21]. The novel multi-tissue, nucleotide-resolution DNA methylation maps of koalas reveal genome-wide patterns of tissue-specific differential DNA methylation enriched in gene bodies. Gene body methylation is an ancestral form of DNA methylation in animal genomes (e.g. [32,34]). Although its role in gene expression has been historically less appreciated than promoter DNA methylation has, gene body DNA methylation is becoming recognized as an important component of transcriptional regulation. For example, a study of human epigenome of 18 tissues reported that differential methylation occurring within gene bodies was more strongly associated with gene expression than those in promoters [23]. Our results indicate that gene body DNA methylation plays similarly significant roles in koala gene regulation.

Studies from other taxa have also demonstrated that the relationship between gene body DNA methylation and gene expression is nonlinear. For example, DNA methylation levels of the first exons/introns of genes are negatively correlated with gene expression [35–37], and tend to be different

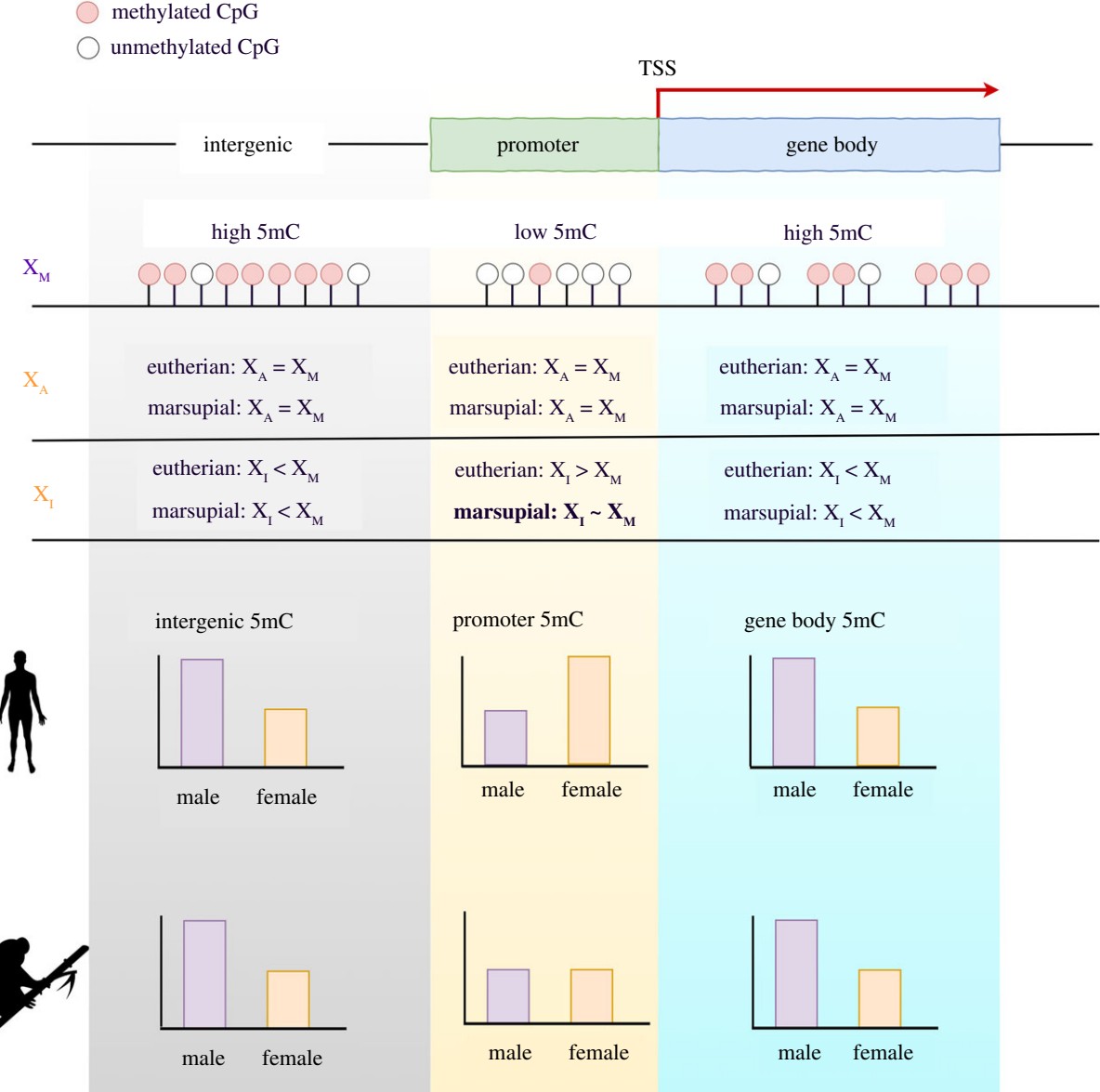

**Figure 5.** Model of DNA methylation (5mC) patterns for representative eutherian and marsupial mammals. In female eutherian mammals, DNA methylation of promoters and CpG islands are increased on the inactive X chromosome ($X_I$) compared to the active X chromosome ($X_A$). In comparison, gene body and intergenic DNA methylation is reduced on the inactive X chromosome ($X_I$) compared to the active X chromosome ($X_A$). Female marsupial mammals show hypomethylation in gene bodies and intergenic regions of the inactive X chromosome; however, they diverge from eutherian mammals in their promoter methylation patterns. Marsupial promoters are modestly hypomethylated in the female X chromosomes ($X_A$ and $X_I$) compared to the male X chromosome ($X_M$).

from downstream genic regions [36]. Conversely, high levels of cumulative gene body DNA methylation are positively correlated with gene expression and may reduce spurious transcription of intragenic RNA [38,39]. The relationship between gene body DNA methylation and transcription in koalas (electronic supplementary material, figure S3B) shows a similar pattern to the observations in other taxa [29–32].

At the chromosomal level, we show that the female X chromosomes of koala are globally hypomethylated compared to both the male X chromosome and the autosomes of both sexes (figure 2). Even though it may appear counterintuitive at the first glance, we posit that the hypomethylation of female X chromosome is as a common feature of eutherian and marsupial mammals driven by the DNA methylation patterns of gene bodies and intergenic regions. Hellman & Chess [40] showed that the inactive X chromosomes of humans had reduced gene body DNA methylation. Whole-genome bisulfite sequencing data of mouse [24] and humans [41] also showed pervasive hypomethylation of

the inactive X chromosome in gene bodies and intergenic regions (figure 2). We present a model summarizing these observations (figure 5).

By contrast, Waters et al. [19] recently proposed that the reduction of gene body DNA methylation was specific to marsupials, but not observed in mouse [19]. The reason why they did not observe DNA methylation difference in mouse might be due to the inherent bias of their method, RRBS, which disproportionately samples regions with high GC content [20]. High GC-content regions tend to be hypomethylated [42,43] and show less variation of DNA methylation. We illustrate this trend using koala data in electronic supplementary material, figure S8. Since RRBS samples high GC genomic regions, the difference between male and female X chromosomes could have been underestimated in the previous study [19]. We also note that since promoters are generally high in GC contents, they show comparatively lower methylation difference between the male and female X chromosomes (electronic supplementary material, figure S8). The causative

relationship between chromosome-wide DNA hypomethylation of the X chromosome and chromosome-wide gene silencing is currently unresolved. Interestingly, marsupial genomes harbour an additional copy of *DNMT1* [44], which could lead to functional divergence between the mammalian lineages. Analyses of DMNT expression in our data, however, did not indicate significant differential expression of *DNMT*s between sexes (probability of differential expression using NOISeq less than 95%).

Despite overarching promoter patterns, DNA methylation signatures of *Rsx*, the major player in XCI initiation in marsupials [8], suggest that koala *Rsx* expression is regulated by DNA methylation of upstream CpG islands (figure 4). Previously, Wang *et al.* [10] showed differential DNA methylation of *Rsx* promoter in opossum. Our observation is consistent with Wang *et al.* [10] and suggests that the regulation of the key initiator of XCI via differential DNA methylation of regulatory sequences is a common feature of eutherians and marsupials.

In summary, we show that gene body DNA methylation is an important contributor to differential expression between tissues in koalas. We also show that the global hypomethylation of female X chromosome (specifically in gene bodies and intergenic regions) is a conserved feature of X chromosome regulation in eutherians and marsupials (figure 5). However, X chromosome promoter methylation and the subsequent effect on the regulation of gene expression appear to be divergent between these two lineages (figure 5). The regulation of the *Rsx*, on the other hand, is supported by promoter DNA methylation, which mirrors the regulation of the eutherian *Xist* locus. Together, these conclusions illuminate the intricate evolutionary pathways that have diverged and converged to influence gene regulation, XCI, and dosage compensation in eutherian and marsupial mammals.

# 4. Methods

## (a) Whole-genome bisulfite sequencing and processing
Genomic DNA was extracted using a Bioline Isolate II Genomic DNA Extraction Kit (cat. no. BIO-52067) following the recommended protocol with an additional DNAse free RNaseA (100 mg ml$^{-1}$) (Qiagen cat. no. 19101) treatment before column purification. Twenty milligram tissue samples from the brain, kidney, lung, skeletal muscle and pancreas from a female koala, 'Pacific Chocolate' (Australian Museum registration M.45022), and a male koala, 'Ben' (Australian Museum registration M.47723), were bisulfite converted using the EX DNA Methylation-Lightning Kit (Zymo cat. no. D5030). WGBS libraries were constructed using the TruSeq DNA methylation kit (Illumina cat. no. EGMK81213). The libraries were sequenced on a NovaSeq6000 S2 (Illumina) using the $2 \times 100$ bp PE option. Processing of the WGBS data followed previous studies [25]. Bisulfite conversion rates were estimated for each WGBS sample using methPipe's bsrate [45] (electronic supplementary material, table S1). Strand-specific methylation calls were combined, and all samples were filtered to remove CpGs covered by fewer than three reads (electronic supplementary material, table S1).

## (b) Analyses of tissue differentially methylated regions
A hierarchical clustering tree was drawn using the *hclust* from R's stats package. The distance matrix was calculated using Euclidean distances and Ward's method was used for the agglomeration. The data for the final tree were visualized using R's dendextend package [46]. Clustering confidence values were generated by pvclust using 10 000 bootstraps. Bismark generated CpG reports were filtered to remove scaffolds that were less than 2 Mb in length, retaining $3.03 \times 10^9$ (94.8%) of the genome. DMRs were called using BSmooth [26], with a minimum fractional methylation difference of 0.3 (30%) and at least 5 CpG sites per DMR. DMRs were considered shared between tissues if they overlapped by at least 50%. Using koala gene annotations from Ensembl (Phascolarctos_cinereus.phaCin_unsw_v4.1.97 release), promoters were defined as regions located 1000 bp upstream of the identified transcription start site (TSS). We generated 10 000 genomic control regions (length and GC content matched) for all unique DMRs for enrichment analyses. Functional annotation and GO term enrichment analysis was performed using the ToppGene Suite [47]. The gene sets were combined for the lung and kidney due to the similarity of their methylation profiles and lack of DMRs (figure 1*a,c*).

## (c) Differential DNA methylation between sexes
We randomly sampled a subset of the autosomal scaffolds that were length matched with the X chromosome scaffolds, which we called the 'matched autosome' dataset. These scaffolds were divided into 10 kb bins and the difference between male and female fractional methylation at each 10 kb bin was computed for all tissues. For the analysis of human data, we used WGBS fractional methylation reports from a male brain (Epigenome ID: E071) and a female brain (Epigenome ID: E053) and the human known gene annotations from Ensembl (hg19 release). Due to its similarity in size to the human X chromosome, we used data from human chromosome 8 as our representative autosome in the comparative analysis. The mean methylation across functional regions was calculated by dividing each gene's function regions into 20 even bins by sequence length. Significance for each bin (Mann–Whitney test) is shown in electronic supplementary material, figure S6.

## (d) Identification of candidate X-linked scaffolds
To isolate candidate X-linked scaffolds from the 1477 unclassified koala scaffolds, we binned the unclassified scaffolds into 10 kb windows and calculated the mean fractional methylation of the associated CpGs. We then determined the average female and male methylation differences across the bins and plotted the density of the differences for all five tissues. SVY and DSi independently select scaffolds that exhibited a statistically significant shift towards female hypomethylation from zero. The scaffolds that showed significant female hypomethylation in all five tissues and were selected by both SVY and DSi were isolated ($n = 98$ covering 14.6 Mb of sequence with mean female–male $5mC = -0.25 \pm 0.12$). As an additional validation, the per cent of reads mapping to the putative X-linked and autosome-linked scaffolds over the total number of mapped reads was computed for the male and female sample in all tissues.

## (e) Annotation of the koala *Rsx*
For *Rsx* annotation, we downloaded the published genome *Rsx* fasta files from the partial opossum assembly [8] and the complete PacBio koala assembly [21,48]. We used BLASTN 2.2.29 [49] to align both sequences to the koala reference genome (phaCin_unsw_v4.1) and obtained genomic coordinates. The entire assembled koala *Rsx* sequence aligned with 100% identity and no gaps. Only one 30.4 kb transcript, a novel *lnc*RNA, overlapped with the annotated *Rsx* region (overlap greater than 90% of transcript) and was used to evaluate gene expression.

## (f) Analysis of differential gene expression

All RNA-seq expression data were obtained from previously published koala transcriptomes [28]. Following the protocol outlined in [50], we used the koala GTF annotation from Ensembl (Phascolarctos_cinereus.phaCin_unsw_v4.1.97.gtf.gz release) to assemble mapped reads into transcripts using StringTie 2.0 [50] with the -e-b–A < gene_abund.tab > flags. We used StringTie's functionality for de novo transcript assembly to identify candidate *Rsx* transcripts. An updated GTF annotation was generated including novel transcripts using the –merge flag and the previously generated mapped reads were reassembled into transcripts guided by this GTF file. DeSeq2 1.22.2 [51] was used to perform differential gene expression analysis between males and females. NOISeq 2.26.1 [33] was used for differential expression analysis due to its ability to simulate technical replicates within given RNA-seq datasets when no replicates are available.

**Data accessibility.** The raw and processed methylation datasets generated in this study have been deposited and accessible through GEO Series accession number GSE149600.

**Authors' contributions.** D.Si., D.A.-P. and S.V.Y. formed the research design; D.A.-P., A.G.K., D.E.A.-P., and R.N.J. generated the data; D.Si., D.Su. and S.V.Y. performed the analysis and wrote the initial draft; all authors contributed to and approved the final manuscript.

**Competing interests.** The authors declare that they have no competing interests.

**Funding.** This work was supported by a grant from the National Science Foundation (MCB 1818288) and a Pilot Grant from the Smooth Muscle Plasticity COBRE of the University of Nevada, Reno (funded by the National Institutes of Health grant 5P30GM110767-04) to D.A.-P., grants by the National Science Foundation (MCB 1615664) and the National Institute of Health (R01MH103517) to S.V.Y. D.Si. was partially supported by the NIH Training Grant in Computational Biology and Biomedical Genomics (T32 GM105490).

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
