## [Peer Review File · Proceedings of the Royal Society B: Biological Sciences]

Review History

RSPB-2020-2244.R0 (Original submission)

Review form: Reviewer 1

Recommendation

Accept with minor revision (please list in comments)

Scientific importance: Is the manuscript an original and important contribution to its field?

Good

General interest: Is the paper of sufficient general interest?

Good

Quality of the paper: Is the overall quality of the paper suitable?

Good

Is the length of the paper justified?

Yes

Should the paper be seen by a specialist statistical reviewer?

No

Do you have any concerns about statistical analyses in this paper? If so, please specify them explicitly in your report.

No

It is a condition of publication that authors make their supporting data, code and materials available - either as supplementary material or hosted in an external repository. Please rate, if applicable, the supporting data on the following criteria.

Is it accessible?

Yes

Is it clear?

Yes

Is it adequate?

Yes

Do you have any ethical concerns with this paper?

No

Comments to the Author

The manuscript entitled « Divergent DNA methylation signatures of X chromosome regulation in marsupials and eutherians » describes the analysis of DNA methylation profiles along the X-chromosome in five different tissues taken from a male and a female Koala. The male/female comparison allows the authors to draw conclusions on the contribution of DNA methylation in the regulation of the inactive state which characterises the paternal X chromosome in females. An important discovery reported here is the lack of CpG hypermethylation at X chromosome promoters in female vs male koala adult tissues. This observation suggests that, unlike the situation at the eutherian inactive X, CpG methylation at koala's X-linked promoters is not as crucial for X silencing as in eutherian mammals. This also correlates with increased expression from the X-chromosome in females compared to males suggesting a more relaxed form of X-inactivation in the koala species.

Major concerns

Overall the study is very descriptive and lacks conclusions at the end of each section. This makes the logical development of the results part very hard to follow. Introducing the question that is addressed in each kind of analysis would be helpful as well.

Relative to Fig1:

- Why are there less DMRs in brain tissues? Technical artefact or reality? How does it affect the results and the statistical analyses?
- Data in sup fig1 and in sup table3 seem to be the same?
- In supplementary fig 3A, very few gene promoters are actually supporting the correlation and some promoters do not follow the trend. This should be mentioned and interpretation should be nuanced. In panel sup 3B, the strongest correlation is between CpG hypomethylation and transcription.

Relative to Fig2:

- Could it be possible to apply stats to data from panels D, E and F. This would support the differential of CpG methylation at X promoters between koalas and humans.
- Panel 2G is not described.
- Yellow line in panel E is not visible.
- There are two supplementary figs 7

- The RNA-seq analysis and the correlation between expression and methylation for genes on the X chr between male and female koala should be shown as main figures. Compile in a fig results from sup 6 panel A and sup 7 panel A & C.

I would change or reformulate the title of the manuscript to replace marsupial by koalas since it is not clear, at this stage, whether observations made here can be extrapolated to the marsupial evolutionary subclade.

Review form: Reviewer 2

Recommendation

Accept with minor revision (please list in comments)

Scientific importance: Is the manuscript an original and important contribution to its field?

Good

General interest: Is the paper of sufficient general interest?

Good

Quality of the paper: Is the overall quality of the paper suitable?

Good

Is the length of the paper justified?

Yes

Should the paper be seen by a specialist statistical reviewer?

No

Do you have any concerns about statistical analyses in this paper? If so, please specify them explicitly in your report.

No

It is a condition of publication that authors make their supporting data, code and materials available - either as supplementary material or hosted in an external repository. Please rate, if applicable, the supporting data on the following criteria.

Is it accessible?

No

Is it clear?

No

Is it adequate?

No

Do you have any ethical concerns with this paper?

No

Comments to the Author

The authors used whole genome bisulfite sequencing to analyze DNA methylation patterns in a female and a male of an unusual marsupial model, the koala. They dissected DNA patterns in 5 different tissues, and made comparison between tissues, and gender. They conclude that gene body DNA methylation seems to be an important contributor to differential gene expression

between tissues in koala. Moreover, they focused on the X chromosome and conclude that, while low methylation of gene bodies of the female X chromosomes is a conserved feature of X chromosome regulation over eutherians and marsupials, promoter DNA methylation patterns are divergent. This study is important for the fields of mammalian evolution, DNA methylation and epigenetics, and X chromosome inactivation. However, some conclusions need more comments and clearer comparisons between marsupial species and eutherians. The article, including the title, focuses on the X chromosome, but it may be considered to be more general on DNA methylation of the whole genome.

Introduction

1) Line 45-47 (sentence « Third, while eutherian XCI... ») : the sentence is not clear. The authors state that the profile of active and repressive histone marks is not consistent on the marsupial inactive X chromosome. In fact, although indeed the profile of enrichment of inactive marks is not consistent between species and/or tissues, the profile of exclusion of active histone marks seems to be consistent on the marsupial inactive X chromosome (refs Koina et al 2009, Rens et al 2010, Chaumeil et al PLOS One 2011).

2) Introduction on DNA methylation: the authors summarize the published results of DNA methylation studies in marsupials and argue that the differences in techniques (immunostaining vs bisulfite sequencing) or the methodologies were not optimal get clear results. I should be noted also that these studies (Wang et al, Ingles and Deakin, Rens et al, Piper and Cooper, Loebel and Johnston, Waters et al) used different marsupial species and different tissues. The authors should comment on it.

Results

- First section (genome-wide patterns...)

3) Related to the previous point, could the authors comment on the choice of the koala to perform their study? Although it is obviously interesting to understand what is happening in more and more species, the use of this unusual model precludes from comparisons with previous studies.

4) While the authors extensively comment on differences of DNA methylation patterns between tissues, they don't comment between male and female. There is one sentence (line 131) but I think it should be commented well before it. Related to it, is the use of 1 male and 1 female only enough to draw conclusions?

- Second section (Differential DNA methylation between tissues)

5) The authors found that pancreas DNA methylation patterns are the most different compared to the others tissues. Is it similar to other marsupial / eutherian species?

6) Again, is the finding of the majority of tissue-specific DMRs being in gene body similar to other species?

7) Same for the correlation DNA methylation/gene expression?

8) Figure 1C is unclear to me. The legend states "Enrichments of DMRs in each functional regions compared to control regions" but the figure shows only the brain. What are the control regions?

9) Minor point: in the legend of Figure 1, "(D) ALDOC..." should be replaced by "(E) ALDOC..."

- Third section (Global hypomethylation of female X...)

10) The authors compared X chromosome DNA methylation of the female koala with the male X and the female autosomes. Is X chromosome DNA methylation of the male lower than autosomal DNA methylation? In other words is it a general property of the X to be hypomethylated, independently of the gender?

11) Is this hypomethylation of the female X specific to 1 of the 2 X chromosomes or is it general to the 2 female X? Although the resolution might not be sufficient, an immunostaining for DNA methylation may show differences between the 2 X chromosomes.

12) Minor point: it seems that the Figure 2G is not called in the main text.

- Fourth section (Promoter DNA methylation...)

13) I am confused and I probably didn't understand properly this section, but it seems that there are more overexpressed X-linked genes in females than in males. The authors may comment on the X chromosome inactivation (XCI) process in koalas: does it occur? Why female X-linked genes are overexpressed? Do they escape XCI? Some RNA FISH experiments may help to assess

whether only one chromosome is expressed.

- 14) Female and male X chromosome expression levels should be compared with other species.
- 15) The Supplementary Fig 7 is confusing to me. In the tables, I don't understand what are the "total genes", and why they are all different.
- 16) Minor point: Some panels of Supplementary Fig 6 and 7 could be shown as a main figure.
 - Fifth section (The R_{sx} region...)
- 17) The data on R_{sx} are interesting, however the XCI status is unclear, so what conclusions can be drawn?
 - Discussion
- 18) The discussion about X chromosome DNA methylation levels is confusing to me (starting line 223). Comparisons between koala and other marsupials should be developed further, as well as between mouse and human. Also distinction between gene body and promoter DNA methylation could be clearer.
- 19) In the discussion the authors state that "gene body DNA methylation is an important contributor to differential expression between tissues" (line 252). In the results, the authors showed that indeed the majority of DMRs are in gene body (line 94), however, they state that the correlation between gene body methylation and gene expression is complex (line 107). Could they comment more on this?
- 20) Related to the comments on the fourth section, the authors could comment more on the X chromosome inactivation process in koalas.
- 21) Minor point: Figure 4: the table is unclear - why are they using X_m, X_i, and X_a?

Decision letter (RSPB-2020-2244.R0)

09-Nov-2020

Dear Professor Yi:

Your manuscript has now been peer reviewed and the reviews have been assessed by an Associate Editor. The reviewers' comments (not including confidential comments to the Editor) and the comments from the Associate Editor are included at the end of this email for your reference. As you will see, the reviewers and the Editors have raised some concerns with your manuscript and we would like to invite you to revise your manuscript to address them.

Research ethics:

Use of animals and field studies:

It is a condition of publication that you make available the data and research materials supporting the results in the article. Please see our Data Sharing Policies (<https://royalsociety.org/journals/authors/author-guidelines/#data>). Datasets should be deposited in an appropriate publicly available repository and details of the associated accession number, link or DOI to the datasets must be included in the Data Accessibility section of the article (<https://royalsociety.org/journals/ethics-policies/data-sharing-mining/>). Reference(s) to datasets should also be included in the reference list of the article with DOIs (where available).

Please submit a copy of your revised paper within three weeks. If we do not hear from you within this time your manuscript will be rejected. If you are unable to meet this deadline please let us know as soon as possible, as we may be able to grant a short extension.

Best wishes,
Dr Locke Rowe
mailto: proceedingsb@royalsociety.org

Associate Editor
Board Member: 1

Comments to Author:

This is an interesting paper that eventually should be published. However, both reviewers make a number of comments related to the presentation of the study, and clarification and discussion of the results. The suggestions for improvement provided should be followed.

Please note the suggestions to change title. I second the idea of replacing marsupials with koala.

Reviewer(s)' Comments to Author:

Referee: 1

Comments to the Author(s)

The manuscript entitled « Divergent DNA methylation signatures of X chromosome regulation in marsupials and eutherians » describes the analysis of DNA methylation profiles along the X-chromosome in five different tissues taken from a male and a female Koala. The male/female comparison allows the authors to draw conclusions on the contribution of DNA methylation in the regulation of the inactive state which characterises the paternal X chromosome in females. An important discovery reported here is the lack of CpG hypermethylation at X chromosome promoters in female vs male koala adult tissues. This observation suggests that, unlike the situation at the eutherian inactive X, CpG methylation at koala's X-linked promoters is not as crucial for X silencing as in eutherian mammals. This also correlates with increased expression from the X-chromosome in females compared to males suggesting a more relaxed form of X-inactivation in the koala species.

Major concerns

Overall the study is very descriptive and lacks conclusions at the end of each section. This makes the logical development of the results part very hard to follow. Introducing the question that is addressed in each kind of analysis would be helpful as well.

Relative to Fig1:

- Why are there less DMRs in brain tissues? Technical artefact or reality? How does it affect the results and the statistical analyses?
- Data in sup fig1 and in sup table3 seem to be the same?
- In supplementary fig 3A, very few gene promoters are actually supporting the correlation and some promoters do not follow the trend. This should be mentioned and interpretation should be nuanced. In panel sup 3B, the strongest correlation is between CpG hypomethylation and transcription.

Relative to Fig2:

- Could it be possible to apply stats to data from panels D, E and F. This would support the differential of CpG methylation at X promoters between koalas and humans.
- Panel 2G is not described.
- Yellow line in panel E is not visible.
- There are two supplementary figs 7
- The RNA-seq analysis and the correlation between expression and methylation for genes on the X chr between male and female koala should be shown as main figures. Compile in a fig results from sup 6 panel A and sup 7 panel A & C.

I would change or reformulate the title of the manuscript to replace marsupial by koalas since it is not clear, at this stage, whether observations made here can be extrapolated to the marsupial evolutionary subclade.

Referee: 2

Comments to the Author(s)

The authors used whole genome bisulfite sequencing to analyze DNA methylation patterns in a female and a male of an unusual marsupial model, the koala. They dissected DNA patterns in 5 different tissues, and made comparison between tissues, and gender. They conclude that gene body DNA methylation seems to be an important contributor to differential gene expression between tissues in koala. Moreover, they focused on the X chromosome and conclude that, while low methylation of gene bodies of the female X chromosomes is a conserved feature of X chromosome regulation over eutherians and marsupials, promoter DNA methylation patterns are divergent. This study is important for the fields of mammalian evolution, DNA methylation and epigenetics, and X chromosome inactivation. However, some conclusions need more comments and clearer comparisons between marsupial species and eutherians. The article, including the title, focuses on the X chromosome, but it may be considered to be more general on DNA methylation of the whole genome.

Introduction

1) Line 45-47 (sentence « Third, while eutherian XCI...): the sentence is not clear. The authors state that the profile of active and repressive histone marks is not consistent on the marsupial inactive X chromosome. In fact, although indeed the profile of enrichment of inactive marks is not consistent between species and/or tissues, the profile of exclusion of active histone marks seems to be consistent on the marsupial inactive X chromosome (refs Koina et al 2009, Rens et al 2010, Chaumeil et al PLOS One 2011).

2) Introduction on DNA methylation: the authors summarize the published results of DNA methylation studies in marsupials and argue that the differences in techniques (immunostaining vs bisulfite sequencing) or the methodologies were not optimal get clear results. I should be noted also that these studies (Wang et al, Ingles and Deakin, Rens et al, Piper and Cooper, Loebel and Johnston, Waters et al) used different marsupial species and different tissues. The authors should comment on it.

Results

- First section (genome-wide patterns...)

3) Related to the previous point, could the authors comment on the choice of the koala to perform their study? Although it is obviously interesting to understand what is happening in more and more species, the use of this unusual model precludes from comparisons with previous studies.

4) While the authors extensively comment on differences of DNA methylation patterns between tissues, they don't comment between male and female. There is one sentence (line 131) but I think it should be commented well before it. Related to it, is the use of 1 male and 1 female only enough to draw conclusions?

- Second section (Differential DNA methylation between tissues)

5) The authors found that pancreas DNA methylation patterns are the most different compared to the others tissues. Is it similar to other marsupial / eutherian species?

- 6) Again, is the finding of the majority of tissue-specific DMRs being in gene body similar to other species?
- 7) Same for the correlation DNA methylation/gene expression?
- 8) Figure 1C is unclear to me. The legend states "Enrichments of DMRs in each functional regions compared to control regions" but the figure shows only the brain. What are the control regions?
- 9) Minor point: in the legend of Figure 1, "(D) ALDOC..." should be replaced by "(E) ALDOC..."
- Third section (Global hypomethylation of female X...)
- 10) The authors compared X chromosome DNA methylation of the female koala with the male X and the female autosomes. Is X chromosome DNA methylation of the male lower than autosomal DNA methylation? In other words is it a general property of the X to be hypomethylated, independently of the gender?
- 11) Is this hypomethylation of the female X specific to 1 of the 2 X chromosomes or is it general to the 2 female X? Although the resolution might not be sufficient, an immunostaining for DNA methylation may show differences between the 2 X chromosomes.
- 12) Minor point: it seems that the Figure 2G is not called in the main text.
- Fourth section (Promoter DNA methylation...)
- 13) I am confused and I probably didn't understand properly this section, but it seems that there are more overexpressed X-linked genes in females than in males. The authors may comment on the X chromosome inactivation (XCI) process in koalas: does it occur? Why female X-linked genes are overexpressed? Do they escape XCI? Some RNA FISH experiments may help to assess whether only one chromosome is expressed.
- 14) Female and male X chromosome expression levels should be compared with other species.
- 15) The Supplementary Fig 7 is confusing to me. In the tables, I don't understand what are the "total genes", and why they are all different.
- 16) Minor point: Some panels of Supplementary Fig 6 and 7 could be shown as a main figure.
- Fifth section (The Rsx region...)
- 17) The data on Rsx are interesting, however the XCI status is unclear, so what conclusions can be drawn?
- Discussion
- 18) The discussion about X chromosome DNA methylation levels is confusing to me (starting line 223). Comparisons between koala and other marsupials should be developed further, as well as between mouse and human. Also distinction between gene body and promoter DNA methylation could be clearer.
- 19) In the discussion the authors state that "gene body DNA methylation is an important contributor to differential expression between tissues" (line 252). In the results, the authors showed that indeed the majority of DMRs are in gene body (line 94), however, they state that the correlation between gene body methylation and gene expression is complex (line 107). Could they comment more on this?
- 20) Related to the comments on the fourth section, the authors could comment more on the X chromosome inactivation process in koalas.
- 21) Minor point: Figure 4: the table is unclear - why are they using Xm, Xi, and Xa?

Author's Response to Decision Letter for (RSPB-2020-2244.R0)

See Appendix A.

RSPB-2020-2244.R1 (Revision)

Review form: Reviewer 1

Recommendation

Accept as is

Scientific importance: Is the manuscript an original and important contribution to its field?

Good

General interest: Is the paper of sufficient general interest?

Good

Quality of the paper: Is the overall quality of the paper suitable?

Good

Is the length of the paper justified?

Yes

Should the paper be seen by a specialist statistical reviewer?

No

Do you have any concerns about statistical analyses in this paper? If so, please specify them explicitly in your report.

No

It is a condition of publication that authors make their supporting data, code and materials available - either as supplementary material or hosted in an external repository. Please rate, if applicable, the supporting data on the following criteria.

Is it accessible?

Yes

Is it clear?

Yes

Is it adequate?

Yes

Do you have any ethical concerns with this paper?

No

Comments to the Author

The authors have appropriately addressed my concerns.

Review form: Reviewer 2

Recommendation

Accept as is

Scientific importance: Is the manuscript an original and important contribution to its field?

Excellent

General interest: Is the paper of sufficient general interest?

Excellent

Quality of the paper: Is the overall quality of the paper suitable?

Excellent

Is the length of the paper justified?

Yes

Should the paper be seen by a specialist statistical reviewer?

No

Do you have any concerns about statistical analyses in this paper? If so, please specify them explicitly in your report.

No

It is a condition of publication that authors make their supporting data, code and materials available - either as supplementary material or hosted in an external repository. Please rate, if applicable, the supporting data on the following criteria.

Is it accessible?

No

Is it clear?

N/A

Is it adequate?

N/A

Do you have any ethical concerns with this paper?

No

Comments to the Author

The major concerns have been addressed and the manuscript has been substantially improved.

Decision letter (RSPB-2020-2244.R1)

26-Jan-2021

Dear Professor Yi

I am pleased to inform you that your manuscript entitled "Koala methylomes reveal divergent and conserved DNA methylation signatures of X chromosome regulation" has been accepted for publication in Proceedings B.

Open Access

Your article has been estimated as being 9 pages long. Our Production Office will be able to confirm the exact length at proof stage.

Paper charges

Sincerely,

Dr Locke Rowe

Appendix A

Associate Editor

Board Member: 1

Comments to Author:

This is an interesting paper that eventually should be published. However, both reviewers make a number of comments related to the presentation of the study, and clarification and discussion of the results. The suggestions for improvement provided should be followed.

Please note the suggestions to change title. I second the idea of replacing marsupials with koala.

Response: Thank you for your thoughtful comments. We have addressed all concerns by the reviewers below and changed the title as suggested to include the koala.

Reviewer(s)' Comments to Author:

Referee: 1

Comments to the Author(s)

The manuscript entitled « Divergent DNA methylation signatures of X chromosome regulation in marsupials and eutherians » describes the analysis of DNA methylation profiles along the X-chromosome in five different tissues taken from a male and a female Koala. The male/female comparison allows the authors to draw conclusions on the contribution of DNA methylation in the regulation of the inactive state which characterizes the paternal X chromosome in females. An important discovery reported here is the lack of CpG hypermethylation at X chromosome promoters in female vs male koala adult tissues. This observation suggests that, unlike the situation at the eutherian inactive X, CpG methylation at koala's X-linked promoters is not as crucial for X silencing as in eutherian mammals. This also correlates with increased expression from the X-chromosome in females compared to males suggesting a more relaxed form of X-inactivation in the koala species.

Major concerns

Overall the study is very descriptive and lacks conclusions at the end of each section. This makes the logical development of the results part very hard to follow. Introducing the question that is addressed in each kind of analysis would be helpful as well.

Response: We have extensively revised the manuscript to clearly state our main questions and motivations for each section. We have also edited the discussion for clarity.

Relative to Fig1:

- Why are there less DMRs in brain tissues? Technical artefact or reality? How does it affect the results and the statistical analyses?

Response: The total numbers of DMRs per tissue are similar between the pancreas and the brain (24089 vs. 20405). The numbers of DMRs in skeletal muscle, kidney and lung are smaller (17475, 13803, and 11333, respectively).

The number of tissue-specific DMRs varies depending on how unique the global methylation profile (measured by the fractional methylation at every CpG) of each tissue is compared to that of all other tissues. The lung and kidney methylomes were the most similar to each other (see Figure 1a) and consequently both lung and kidney samples have the fewest unique, tissue-specific DMRs (22 and 119, respectively). Conversely, the koala pancreas exhibited the most unique methylome in terms of global fractional methylation patterns and consequently had the greatest number of unique, tissue-specific DMRs. Further analysis with a more comprehensive sampling of tissues is required to determine if the pancreas is a true outlier in terms of DNA methylation in this species. However, it is worthwhile to note that koalas are known for their unique and highly specialized diet of eucalyptus leaves, which is highly toxic to most other mammals (1). We added this point in the revised manuscript.

To demonstrate that our observation is not a technical artifact, we additionally performed bootstrapping using *pvclust* (10,000 bootstraps) on the hierarchical clustering analysis and reported associated confidence values for statistical support. In the revised manuscript, we mention the numbers of total DMRs for pancreas and brain in the text. We also updated Fig. 1a to include the bootstrapping support, and Fig. 1b to include the total number of DMRs per tissue (pancreas: 24089, brain: 20405, skeletal muscle: 17475, kidney: 13803, lung: 11333).

- Data in sup fig1 and in sup table3 seem to be the same?

Response: These two supplements do present the same data, however supplementary figure 1 visualizes the relative enrichment of DMRs in different functional regions compared to control regions while supplementary table 3 additionally reports the enrichment values as well as the breakdown of the percent of DMRs in each functional category per tissue. We considered that providing both resources in the supplementary section is preferable for transparency of our interpretations.

- In supplementary fig 3A, very few gene promoters are actually supporting the correlation and some promoters do not follow the trend. This should be mentioned and interpretation should be nuanced. In panel sup 3B, the strongest correlation is between CpG hypomethylation and transcription.

Response: We revised the text to present more nuanced interpretations of these findings. We also added texts in the discussion to this point.

Relative to Fig2:

- Could it be possible to apply stats to data from panels D, E and F. This would support the differential of CpG methylation at X promoters between koalas and humans.

Response: Additional figure, supplementary figure 6, has been added to show the associated p-values for panels D, E, and F in Figure 2.

- Panel 2G is not described.

Response: The panel is now referenced in the main text.

- Yellow line in panel E is not visible.

Response: The yellow line is directly under the purple line in Figure 2E which is expected because, on average, there should be no global sex-based difference between fractional methylation on autosomes. An additional comment has been added in the figure legend to clarify this point. The associated p-values are also reported in the new supplementary figure 6.

- There are two supplementary figs 7

Response: Fixed.

- The RNA-seq analysis and the correlation between expression and methylation for genes on the X chr between male and female koala should be shown as main figures. Compile in a fig results from sup 6 panel A and sup 7 panel A & C.

Response: Following the reviewer's suggestion, we added a new main figure (Figure 3).

I would change or reformulate the title of the manuscript to replace marsupial by koalas since it is not clear, at this stage, whether observations made here can be extrapolated to the marsupial evolutionary subclade.

Response: We changed the title to "Koala methylomes reveal divergent and conserved DNA methylation signatures of X chromosome regulation"

Referee: 2

Comments to the Author(s)

The authors used whole genome bisulfite sequencing to analyze DNA methylation patterns in a female and a male of an unusual marsupial model, the koala. They dissected DNA patterns in 5 different tissues, and made comparison between tissues, and gender. They conclude that gene body DNA methylation seems to be an important contributor to differential gene expression between tissues in koala. Moreover, they focused on the X chromosome and conclude that, while low methylation of gene bodies of the female X chromosomes is a conserved feature of X chromosome regulation over eutherians and marsupials, promoter DNA methylation patterns are divergent. This study is important for the fields of mammalian evolution, DNA methylation and epigenetics, and X chromosome inactivation. However, some conclusions need more comments and clearer comparisons between marsupial species and eutherians. The article, including the title, focuses on the X chromosome, but it may be considered to be more general on DNA methylation of the whole genome.

Response: Response: We changed the title to “Koala methylomes reveal divergent and conserved DNA methylation signatures of X chromosome regulation”

Introduction

1) Line 45-47 (sentence « Third, while eutherian XCI...): the sentence is not clear. The authors state that the profile of active and repressive histone marks is not consistent on the marsupial inactive X chromosome. In fact, although indeed the profile of enrichment of inactive marks is not consistent between species and/or tissues, the profile of exclusion of active histone marks seems to be consistent on the marsupial inactive X chromosome (refs Koina et al 2009, Rens et al 2010, Chaumeil et al PLOS One 2011).

Response: We have modified this sentence to clarify this point and include details.

2) Introduction on DNA methylation: the authors summarize the published results of DNA methylation studies in marsupials and argue that the differences in techniques (immunostaining vs bisulfite sequencing) or the methodologies were not optimal get clear results. I should be noted also that these studies (Wang et al, Ingles and Deakin, Rens et al, Piper and Cooper, Loebel and Johnston, Waters et al) used different marsupial species and different tissues. The authors should comment on it.

Response: We modified the text following the reviewer’s suggestion.

Results

- First section (genome-wide patterns...)

3) Related to the previous point, could the authors comment on the choice of the koala to perform their study? Although it is obviously interesting to understand what is happening in

more and more species, the use of this unusual model precludes from comparisons with previous studies.

Response: The reference assembly generated by Johnson et al. utilizing PacBio long read sequencing is the highest quality reference genome of any marsupial species (2) which was a feature we were interested in leveraging with our whole-genome methylation data. Modern koalas represent the only extant member of the marsupial family Phascolarctidae, a divergent group of marsupials. Of the two marsupial clades, only the methylomes of *Monodelphis*, which excludes Australian marsupials, have been studied thus far (3). Therefore, this species provides another comparison point to integrate previously generated marsupial data.

4) While the authors extensively comment on differences of DNA methylation patterns between tissues, they don't comment between male and female. There is one sentence (line 131) but I think it should be commented well before it. Related to it, is the use of 1 male and 1 female only enough to draw conclusions?

Response: As the reviewer noted, we felt that the dataset, which is one of the first of its kind in marsupials, is not well suited to examine differences between males and females per tissue across the whole genome given that there is only one male and one female sample. Instead, we focused on comparing DNA methylation between tissues which have been shown in previous studies of human, mouse and other marsupial species to exhibit drastic differences that can be identified from single sample analyses (e.g. (4)). We also focused on the differences between male and female X chromosomes, which can also be detected using single samples (e.g., (5)), particularly given that we have replicate data from five tissues. We hope to increase our sample size and examine methylation and gene expression differences between males and females across the genome in future studies.

- Second section (Differential DNA methylation between tissues)

5) The authors found that pancreas DNA methylation patterns are the most different compared to the others tissues. Is it similar to other marsupial / eutherian species?

Response: The pancreas was distinct among the five tissues analyzed here. We have provided additional support for the clustering by providing p-values associated with 10,000 bootstraps of the hierarchical clustering (Figure 1A). Whether the pancreas is truly an outlier in terms of DNA methylation must be determined in a future study with a more comprehensive sampling of tissues. Analysis of human tissues showed that reproductive tissues such as placenta and sperm tend to have more distinct DNA methylation patterns compared to somatic tissues (e.g., (4)). Because DNA methylation data from different tissues from other species are currently lacking, we cannot extend our results to other marsupial species. Nevertheless, we note that koalas have a highly specialized diet of eucalyptus leaves which are highly toxic to most other mammals (1). This observation, coupled with the important role of the pancreas in generating digestive enzymes, could support the reported pattern. However, since we only compared five tissues, we

refrained from making a strong conjecture. We included a sentence in the revised results and discussion.

6) Again, is the finding of the majority of tissue-specific DMRs being in gene body similar to other species?

Response: As far as we are aware, tissue specific whole genome methylation maps are rare outside of human and mouse. Data from human tissues also reported DMRs abundant in gene bodies, as we cite in the text (6).

7) Same for the correlation DNA methylation/gene expression?

Response: The correlation between gene body DNA methylation and gene expression is complex, as we have cited in the manuscript. We have provided more details in the revised Results and Discussion sections.

8) Figure 1C is unclear to me. The legend states “Enrichments of DMRs in each functional regions compared to control regions” but the figure shows only the brain. What are the control regions?

Response: The legend has been updated for clarity. The control regions were generated from subsets of genomic regions that matched the length and GC content of each DMR. The figure shows the relative enrichment or depletion of DMRs in each functional region (promoter, gene body, or intergenic region) compared to the randomly selected genomic regions with matched characteristics. For simplicity (and because the patterns were consistent across all five tissues), only one tissue (the brain) was reported in the main figure. Data from other tissues are shown in Supplementary Figure 1 and now noted in the legend.

9) Minor point: in the legend of Figure 1, “(D) ALDOC...” should be replaced by “(E) ALDOC...”

Response: Fixed. In the revision we only show the ALDOC locus DMR for brevity.

- Third section (Global hypomethylation of female X...)

10) The authors compared X chromosome DNA methylation of the female koala with the male X and the female autosomes. Is X chromosome DNA methylation of the male lower than autosomal DNA methylation? In other words is it a general property of the X to be hypomethylated, independently of the gender?

Response: As shown in Figure 2C, the X chromosome is similarly methylated to autosomes in both human and koala males. This point is now emphasized in the Results section.

11) Is this hypomethylation of the female X specific to 1 of the 2 X chromosomes or is it general

to the 2 female X? Although the resolution might not be sufficient, an immunostaining for DNA methylation may show differences between the 2 X chromosomes.

Response: Even though we do not have sufficient resolution, as discussed in the model illustrated in Figure 5, we proposed that this was due to the X chromosome inactivation.

12) Minor point: it seems that the Figure 2G is not called in the main text.

Response: Fixed.

- Fourth section (Promoter DNA methylation...)

13) I am confused and I probably didn't understand properly this section, but it seems that there are more overexpressed X-linked genes in females than in males. The authors may comment on the X chromosome inactivation (XCI) process in koalas: does it occur? Why female X-linked genes are overexpressed? Do they escape XCI? Some RNA FISH experiments may help to assess whether only one chromosome is expressed.

Response: Experiments using isozyme, single nucleotide primer extension, and FISH of candidate genes have shown that one of the X chromosomes appears to be inactivated in several marsupial species (references in the main text). However, genomic and transcriptomic resources from marsupials are still relatively rare, and a comprehensive analysis of genome-wide transcription between males and females (other than comparisons of the X chromosome) appears sparse. We hope to expand to perform such an analysis in the future as the resources develop further.

14) Female and male X chromosome expression levels should be compared with other species.\

Response: Sex-inclusive, multi-tissue matched methylation and expression data, like those presented in this manuscript, are novel in marsupial species making meaningful gene expression normalization and multi-species comparisons limited at this time. Indeed the recent work of Naqvi et al. highlights some of the challenges of such analysis in even well-studied mammalian species with rich transcriptional datasets (7). However, we feel this methylome dataset can be used as a resource to be integrating with future work to address these important questions.

15) The Supplementary Fig 7 is confusing to me. In the tables, I don't understand what are the "total genes", and why they are all different.

Response: The total genes value refers to both the significant and nonsignificant genes in each quadrant. This figure has been modified and moved to the main text (Figure 3). An additional supplementary table has been added with an updated legend for clarity (Supplementary Table 4).

16) Minor point: Some panels of Supplementary Fig 6 and 7 could be shown as a main figure.

- Fifth section (The R_{sx} region...)

Response: New main figure (Figure 3) added which compiles these expression results.

17) The data on *Rsx* are interesting, however the XCI status is unclear, so what conclusions can be drawn?

Response: Previous works have established that XCI is observed in marsupials (8-10) and that the *lncRNA* gene *Rsx*, like its mammalian counterpart *Xist*, plays an essential role in mediating this process (11). What has not been shown, at the resolution that we provide with our dataset, is the DNA methylation profile of all functional regions associated with the gene independent of sequence CpG content. Our identification of a female hypomethylated “promoter” region establishes that the sex-specific expression of *Rsx* in koalas follows the patterns observed in eutherian mammals even though the promoters of most X-linked genes did not exhibit the female hypomethylation associated with DNA methylation mediated regulation.

- Discussion

18) The discussion about X chromosome DNA methylation levels is confusing to me (starting line 223). Comparisons between koala and other marsupials should be developed further, as well as between mouse and human. Also distinction between gene body and promoter DNA methylation could be clearer.

Response: Additional interpretations have been added to the discussion.

19) In the discussion the authors state that “gene body DNA methylation is an important contributor to differential expression between tissues” (line 252). In the results, the authors showed that indeed the majority of DMRs are in gene body (line 94), however, they state that the correlation between gene body methylation and gene expression is complex (line 107). Could they comment more on this?

Response: Additional interpretations have been added to the discussion.

20) Related to the comments on the fourth section, the authors could comment more on the X chromosome inactivation process in koalas.

Response: Additional interpretations have been added to the discussion.

21) Minor point: Figure 4: the table is unclear – why are they using X_m , X_i , and X_a ?

Response: This figure is a schematic summary of the methylation signatures across functional regions of the X chromosomes of male and female koalas. As stated in previous works (5), the methylation patterns of the solitary male X chromosome (denoted X_M as explained in the legend) can be used as a baseline proxy to represent the methylation patterns of the active female X chromosome (X_A) and subsequently aid in distinguishing methylation levels on the inactive

female X chromosome (X_1). The table shows the baseline average methylation signature of the male X chromosome as well as the comparative levels in intergenic regions, promoters, and gene bodies in female X chromosomes (active and inactive) for both marsupials and eutherians. The patterns are conserved between the two mammal lineages in all regions except for the inactive female X chromosome's promoters (denoted in bold). The legend and figure have been updated for clarity.

References

1. Gleadow RM, Haburjak J, Dunn JE, Conn ME, Conn EE. Frequency and distribution of cyanogenic glycosides in *Eucalyptus* L'Hérit. *Phytochemistry*. 2008;69(9):1870-4.
2. Johnson RN, O'Meally D, Chen Z, Etherington GJ, Ho SYW, Nash WJ, et al. Adaptation and conservation insights from the koala genome. *Nature Genetics*. 2018;50(8):1102-11.
3. Duchêne DA, Bragg JG, Duchêne S, Neaves LE, Potter S, Moritz C, et al. Analysis of Phylogenomic Tree Space Resolves Relationships Among Marsupial Families. *Systematic Biology*. 2017;67(3):400-12.
4. Mendizabal I, Shi L, Keller TE, Konopka G, Preuss TM, Hsieh TF, et al. Comparative Methylome Analyses Identify Epigenetic Regulatory Loci of Human Brain Evolution. *Mol Biol Evol*. 2016;33(11):2947-59.
5. Waters SA, Livernois AM, Patel H, O'Meally D, Craig JM, Marshall Graves JA, et al. Landscape of DNA Methylation on the Marsupial X. *Molecular biology and evolution*. 2017;35(2):431-9.
6. Schultz MD, He Y, Whitaker JW, Hariharan M, Mukamel EA, Leung D, et al. Human body epigenome maps reveal noncanonical DNA methylation variation. *Nature*. 2015;523(7559):212-6.
7. Naqvi S, Godfrey AK, Hughes JF, Goodheart ML, Mitchell RN, Page DC. Conservation, acquisition, and functional impact of sex-biased gene expression in mammals. *Science (New York, NY)*. 2019;365(6450):eaaw7317.
8. Shevchenko AI, Zakharova IS, Zakian SM. The evolutionary pathway of x chromosome inactivation in mammals. *Acta Naturae*. 2013;5(2):40-53.
9. Sharman GB. Late DNA Replication in the Paternally Derived X Chromosome of Female Kangaroos. *Nature*. 1971;230(5291):231-2.
10. Wang X, Douglas KC, Vandenberg JL, Clark AG, Samollow PB. Chromosome-wide profiling of X-chromosome inactivation and epigenetic states in fetal brain and placenta of the opossum, *Monodelphis domestica*. *Genome Res*. 2014;24(1):70-83.
11. Grant J, Mahadevaiah SK, Khil P, Sangrithi MN, Royo H, Duckworth J, et al. R_{sx} is a metatherian RNA with Xist-like properties in X-chromosome inactivation. *Nature*. 2012;487(7406):254-8.